# Construct validation and measurement invariance of the Parasocial Relationships in Social Media survey

**Austin T. Boyd[1,2], Louis M. Rocconi[1]*, Jennifer Ann Morrow[1]**

**1** Educational Leadership and Policy Studies, The University of Tennessee, Knoxville, Knoxville, Tennessee, United States of America, **2** American Reading Company, Blue Bell, Pennsylvania, United States of America

* lrocconi@utk.edu

**Data Availability Statement:** Data and code are available in the Open Science Framework at https://osf.io/du7r6/ (DOI 10.17605/OSF.IO/DU7R6).

**Funding:** The authors received no specific funding for this work.

## Abstract

This paper examines the construct validity and measurement invariance of the Parasocial Relationships in Social Media (PRISM) survey which was designed to provide researchers with a valid and reliable tool for measuring parasocial relationships developed in a social media context. A confirmatory factor analysis indicated the survey provides an adequate measure of parasocial relationships with online, social media celebrities, replicating the factor structure found by Boyd and colleagues when they developed PRISM and providing evidence of the construct validity of the survey. Additionally, scalar measurement invariance was achieved which supports the survey's ability to compare parasocial relationships across different social media platforms.

## Introduction

Parasocial relationships are intrinsic to modern media consumption. These one-sided relationships can feel as real to viewers as real-life relationships [1] and can influence one's opinions and behaviors beyond media consumption itself [2–4]. While parasocial relationships were first defined prior to the creation of the internet and social media, they are no less present in these mediums. In fact, social media allows content to be created, uploaded, and interacted with on demand and more frequently than traditional media (e.g., television, radio). This allows people to have more frequent parasocial interactions, which in turn can lead to stronger parasocial relationships developing [5–7].

Although there exist various established surveys for assessing parasocial relationships, such as the Parasocial Interaction Scale (PSI) [8], Audience-Persona Interaction Scale (API) [9], and Celebrity-persona Parasocial Interaction Scale (CPPI) [10], none of these instruments were specifically designed nor validated for application in a social media context. The social media context differs from other media formats in terms of the frequency, immediacy, interactivity, and diversity of interactions that occur on these platforms. All of the existing published surveys were designed for traditional media formats, such as television or radio, and may not capture the nuances and dynamics of the social media context. Moreover, none of these surveys can be used in a social media context without major modifications to the original items

**Competing interests:** The authors have declared that no competing interests exist.

due to their specificity and wording. For example, the Parasocial Interaction Scale [8] primarily focuses on assessing relationships with newscasters and specifically refers to individuals' opinions on news stories. Similarly, the Audience-Persona Interaction Scale [9] revolves around celebrities as actors and refers to the characters they portray. Therefore, the Parasocial Relationship in Social Media (PRISM) survey [11] was carefully developed and validated in order to address this limitation of current instruments.

While having one survey for all media contexts would be ideal, it would be challenging and potentially less accurate and useful. Each media context has its own characteristics that may affect the nature and quality of the parasocial relationships that people form with media personalities or characters. Therefore, each context may need its own survey instrument or at least some modifications or adaptations of an instrument to suit the specific context. For instance, the PRISM survey that was developed for social media use may not be suitable for measuring the parasocial relationships that people form with fictional characters in books or movies, as these characters may not have the same level of realism or interactivity as social media celebrities. Similarly, the PRISM survey may not be applicable to measuring the parasocial relationships that people form with news anchors or political leaders, as these figures may have different roles, functions, and influences than social media celebrities. Thus, creating a scale for use with social media rather than one that could be used for any context allows researchers to focus on the particularities and complexities of this phenomenon and to provide a more reliable and valid measure of the parasocial relationships that people form with social media celebrities.

In addition, several researchers [12, 13] have expressed concerns regarding the insufficient construct validation testing conducted on most parasocial phenomena scales. Moreover, many of the existing scales, including the widely used PSI and API scales, were developed before the differentiation between parasocial relationships and interactions gained prominence in the literature. This has led some scholars to argue that these earlier scales are mislabeled and fail to accurately measure their intended constructs [12–14]. These reasons, coupled with the growing impact and prevalence of online celebrities, prompted the development of the Parasocial Relationships in Social Media (PRISM) Survey to fill this gap in the literature [11]. By continuing to validate the PRISM Survey, we hope to provide researchers with a psychometrically sound measure that can be used to assess the parasocial relationships which people develop with online celebrities and influencers through social media.

## Purpose of study and research questions

The purpose of the current study is to assess the psychometric properties of the Parasocial Relationships in Social Media (PRISM) survey. In this study, we investigate the reliability, construct validity, and measurement invariance of the PRISM Survey. The survey was specifically developed to assess the parasocial relationships people develop with social media celebrities online. There are three research questions guiding the current study.

**1: Does the previously identified four factor structure replicate in a new sample?** This first research question explores whether the previously established four-factor item structure holds in a new sample. We addressed this research question by conducting a confirmatory factor analysis (CFA) on a new, independent data set to confirm that the same items load onto the same factors as found in a previously conducted exploratory factor analysis by Boyd et al. [11]. We assess global and local model fit using the guidelines provided by Brown [15] and Gana and Broc [16].

**2: Do the four factors possess adequate convergent and discriminant validity?** The second research question explores the extent to which items within each factor adequately

measure that factor (convergent validity) and the extent to which factors are distinct from one another (discriminant validity). Both convergent and discriminant validity play crucial roles in establishing the overall construct validity of an instrument by examining the relationships among factors. To assess convergent and discriminant validity within the PRISM survey, we employed standardized factor loadings, average variance extracted for each factor, and correlations between factors. Additionally, we examined convergent and discriminant validity by correlating the PRISM factors with the frequency of respondent interactions with the celebrity's online content and the year in which the respondent first started following the celebrity.

**3: Do the measurement properties (e.g., factor structure, factor loadings, threshold parameters) remain invariant across different social media platforms (e.g., live, recorded)?** To further assess the psychometric integrity of the item set, the final research question examines whether the measurement properties (e.g., factor structure, factor loadings, threshold parameters) of the PRISM survey are invariant across different types of social media platforms. Specifically, we examined whether the measurement properties held when assessing a recorded platform (e.g., YouTube) and a live format (e.g., Twitch). Examining whether the same measurement properties hold in different platforms is essential for making comparisons of parasocial relationships across social media platforms.

## Parasocial relationships

The concept of parasocial interaction was originally introduced by Horton and Wohl [17], who defined it as a form of one-sided relationship that develops between a viewer or spectator and the persona of a performer. It is this one-sidedness that makes parasocial relationships distinctly different from other types of relationships, such as romantic relationships or mentor/mentee relationships, which by definition, must be shared and reciprocated [18, 19]. When originally conceived, television was widely regarded as the primary medium through which parasocial relationships could be formed [17, 20]. However, in the 65 years since, research on parasocial phenomena has expanded beyond television to include radio, literature, sports, politics, and social media, such as Facebook, Twitter/X, and YouTube. Over time, our understanding of parasocial interactions has evolved and become more nuanced. While originally used interchangeably, parasocial interactions and relationships are now viewed as distinct constructs in the literature. In general, the definition of parasocial relationships remains consistent with Horton and Wohl's [17] original formulation, that is a one-sided relationship formed by a viewer with a performer that cannot be reciprocated by the performer. Rihl and Wegener [21] added that these relationships are enduring and cross-situational, extending beyond the initial act of media consumption. Parasocial interactions, on the other hand, have come to be defined as the self-contained interactions that occur between a viewer and a performer, *confined to the duration of the media exposure* [22]. Through repeated instances of parasocial interactions, parasocial relationships have the potential to develop, strengthen, and endure over time [12], even though they are not reciprocated by the performer.

An important part of the literature surrounding parasocial relations has been developing scales to measure parasocial relationships effectively. Liebers and Schramm [23] found that in the 65 years since Horton and Wohl's [17] initial publication, there have been over 260 different articles published on parasocial phenomena, 52.4% of which used surveys for data collection. The two most popular measures used in these publications are the Parasocial Interaction Scale (PSI) [8] and Audience-Persona Interaction Scale (API) [9]. Together these two scales have been used in over 100 different publications. The PSI scale is a 20-item measure that was originally designed to measure the parasocial interactions with TV newscasters. The PSI scale also exists as a more compact 10-item measure that was originally designed to measure the

parasocial interactions with soap opera actors. The API scale consists of 22 items and was originally developed to measure the parasocial interactions developed with sitcom characters. Each of these scales use the same 5-point Likert scale from "strongly disagree" to "strongly agree". While these scales are pervasive in the literature, recent research has begun to question their validity for a number of reasons. First, both scales were created prior to the distinction being made between parasocial relationships and parasocial interactions, and some argue that these scales are now mislabeled as they are actually measures of parasocial relationships and not parasocial interactions [12, 13]. Additionally, the scales have very specific wording that requires them to be modified if they are used outside of their original contexts mentioned above. While modifying the surveys for a new context is an option, the surveys would need to be reassessed to determine if they are still valid for the new setting in order to prevent compromising the results [24]. Even the switch between survey formats (e.g., paper to electronic) can cause validity concerns which should be considered [24, 25]. Dibble et al. [12] also argue that most parasocial relationship scales have not undergone adequate tests of construct validation. In this paper we have further validated the PRISM survey to provide researchers with an appropriate survey for measuring parasocial relationships which does not need to be modified for use in social media contexts, thus mitigating potential validity issues, and increasing the overall quality of parasocial relationship research conducted.

## Parasocial Relationships in Social Media (PRISM) survey

The Parasocial Relationships in Social Media (PRISM) Survey was developed by Boyd et al. [11] to assess the parasocial relationships formed by individuals with online social media celebrities. The PRISM survey comprises 22 items. To create the PRISM survey, Boyd et al. [11] adapted and modified items from three well-established parasocial surveys to suit a social media context. These surveys include the Audience-Persona Interaction Scale [9], Parasocial Interaction Scale [8], and Celebrity-Persona Parasocial Interaction Scale [10]. First, Boyd et al. [11] conducted a thorough comparison to identify relevant and distinctive items from each of the three scales. These items were then modified to align with the specific dynamics and characteristics of parasocial relationships in the realm of social media. The PRISM survey uses a five-point Likert response scale ranging from "Strongly Disagree" to "Strongly Agree" with an additional "Does not Apply" option. Upon completion of the initial item draft, Boyd et al. [11] began the exploration of the PRISM survey's content and construct validity. An external expert review panel was utilized to establish content validity, providing feedback on the content and clarity of the items. This feedback was reviewed and incorporated prior to data collection for the Exploratory Factor Analysis (EFA). The EFA revealed four unique factors that Boyd et al. [11] named "Interest In" (Cronbach's alpha = .83), "Knowledge Of" (Cronbach's alpha = .83), "Identification With" (Cronbach's alpha = .84), and "Interaction With" (Cronbach's alpha = .78) the celebrity of interest consisting of 7, 5, 6, and 4 items, respectively. To further validate these findings, a confirmatory factor analysis was conducted to test the replicability of these four factors in a new sample.

## Methods

### Sampling and recruitment of participants

The study targeted individuals aged 18 years or older who followed online celebrities or content creators on various social media platforms such as Facebook, TikTok, Twitch, and YouTube. Given the wide-ranging population of interest, participant selection did not occur at a centralized location. This facilitated data collection on an international scale using the online survey software platform QuestionPro. The survey was posted across multiple social media

platforms, including Reddit, OnlyFans, Facebook, and LinkedIn. Furthermore, contact was made via email with professors from four universities in the United States, requesting their assistance in distributing the survey to their students. Those who agreed were asked to forward the recruitment materials, along with a link to the online survey, to their students via email.

Participants were recruited from May 2021 to June 2021. Before completing the survey, participants were provided with an informed consent form that outlined the purpose of the study, the potential risks and benefits associated with their participation, and their rights as participants. All participants provided written informed consent by selecting that they had read and understood the information in the informed consent. Data collection was conducted in an anonymous manner, and the research team did not have access to any personally identifiable information of individual participants during or after the data collection process. Ethical approval for the study was obtained from the University of Tennessee Institutional Review Board (UTK IRB-20-06104-XM), ensuring that the study adhered to ethical guidelines and standards for research involving human participants.

## Data screening

To ensure the integrity and quality of the data, several screening measures were implemented. Participants who indicated that they were under 18 years of age, did not follow a celebrity, or did not provide informed consent were automatically excluded by the survey platform. Furthermore, participants who listed multiple different celebrities, celebrities who primarily engage with their audience outside of social media (e.g., actors, musicians), or provided fake names for the celebrities they follow were also removed from the dataset. Furthermore, to ensure consistency, the celebrities listed by participants underwent a recoding process correcting spelling errors, standardizing abbreviations, and aligning pseudonyms or stage names with their corresponding real names. As a result of the data screening process, a final sample size of 606 participants was utilized for further analysis.

## Participants

The data collection phase spanned five weeks, allowing participants to complete the survey at their convenience using a computer, smartphone, or tablet. The age range of participants ranged from 18 to 63 years old with an average age slightly over 24 years old and a standard deviation of 6.7. Among the participants, around 42% identified as female and 42% identified as male, and the remaining participants either identified as another gender identity or did not provide their gender identity. In terms of ethnicity, the majority of participants identified as Caucasian/White (76%). Regarding education, the largest proportion of participants indicated having some college education but no degree (29%), followed by those who had obtained a bachelor's degree (26%). Furthermore, a significant number of participants identified themselves as full-time students (37%), while a quarter reported being employed and working 40 or more hours per week.

In total, participants listed 227 different online celebrities they were following. These online celebrities were provided by the participants when asked to list their single, favorite online celebrity from any social media platform, and then used to populate the name in the participant's survey. When asked how frequently respondents interacted with the celebrity's content, the majority reported engaging with the content either daily (30%) or 3 to 5 days per week (37%). In terms of the year participants began following the celebrity, the most commonly selected years were 2018 (15%), 2019 (20%), and 2020 (19%). The range of years spanned from 2005 to 2021. The primary platforms on which participants followed the celebrity were

YouTube (75%) and Twitch (18%). Additionally, the vast majority of participants (90%) indicated that they followed the celebrity on multiple social media platforms.

## Validity analyses

To address our first research question and further validate the PRISM Survey, a confirmatory factor analysis (CFA) was conducted, specifically targeting the four-factor solution previously identified by Boyd et al. [11] in their exploratory factor analysis. As highlighted by Brown and Moore [26], CFA offers a psychometric evaluation of the underlying structure of the measurement model. Various fit indices were employed to assess the overall goodness of fit of the model, including the comparative fit index (CFI), Tucker-Lewis index (TLI), root mean square error of approximation (RMSEA), and standardized root mean square residual (SRMR). Guidelines proposed by Brown [15] and Gana and Broc [16] were used for interpreting these indices: CFI and TLI values of $\geq .90$ indicate adequate fit, while values of $\geq .95$ indicate good fit; RMSEA values of $\leq .08$ suggest adequate fit, while values of $\leq .05$ indicate good fit; and SRMR values of $\leq .08$ imply adequate fit, while values of $\leq .06$ indicate good fit. Additionally, local fit was assessed by examining residual correlations and modification indices.

To answer our second research question, we assessed the convergent and discriminant validity of the PRISM survey. Using the guidelines suggested by Cheung and Wang [27], we calculated the standardized factor loadings and average variance extracted (AVE) for each factor and calculated the correlations between the factors to assess for convergent and discriminant validity, respectively. Using these guidelines, we can conclude convergent validity when standardized factor loadings and AVE for the factors are greater than 0.50 and discriminant validity when correlations between factors are less than 0.70. Furthermore, we also replicated Boyd et al. [11] and examined the correlations between the PRISM factors and two additional variables: the year in which respondents initially started following the celebrity and a question regarding the frequency of their interactions with the celebrity's online content (e.g., daily, 3 to 5 times per week, once a week, once every two weeks). The use of an existing parasocial relationship survey (e.g., PSI, API) to assess convergent validity was not deemed appropriate since most of the items would need to be revised to fit an online context and several of the items from these scales were adapted and modified during the development of the PRISM survey.

Finally, we address the third research question by examining whether the PRISM survey was invariant across different social media platforms. Specifically, we assessed the measurement invariance between YouTube (i.e., a recorded platform) and Twitch (i.e., a live platform). While measurement invariance between nested models is usually assessed using a chi-squared difference test, research has demonstrated that the chi-squared difference test is influenced by sample size [28–30]. In light of this, Chen [31] suggests that fit indices that are not dependent on sample size, such as CFI and RMSEA, should be used to test measurement invariance between nested models, instead of relying solely on the chi-squared difference test. Using the guidelines offered by Chen [31], $\Delta$CFI $\geq$ -.010 and $\Delta$RMSEA $\geq .015$ indicate non-invariance. However, Chen suggests prioritizing $\Delta$CFI as the complexity of a model can impact $\Delta$RMSEA.

## Results

Given the ordinal nature of the survey items (i.e., Likert scales), we used a diagonally weighted least squares (DWLS) estimation method as it makes no distributional assumptions, such as multivariate normality, and is well suited for use with ordinal response options [15]. A CFA model measuring the four factors found in Boyd et al. [11] (i.e., *Interest In*, *Knowledge Of*, *Identification With*, and *Interaction With)* was examined using the lavaan package [32] in R [33]. Using the guidelines set by Brown [15] and Gana and Broc [16], we determined that the model

displayed adequate fit, CFI = .936, TLI = .927, RMSEA = .073, 90% CI$_{RMSEA}$ (.068, .078), SRMR = .07. While the fit indices could be improved to meet more stringent guidelines by making modifications to the model, such as allowing an error covariance between "Identification with" items 1 and 2 and"Interest in" items 1 and 2, we decided the model fit was adequate and stayed true to the theoretical model we were testing. Although these values do not meet the stricter benchmarks set by Hu and Bentler [34], more recent research has shown that these benchmarks are too stringent, and even Bentler's [35] original CFI and TLI benchmarks of .9 are overly demanding [36, 37]. These results support the four-factor solution found Boyd's [11] in the previously conducted EFA, addressing research question 1. Standardized factor loadings ranged from .56 to .81 for the *Interest in* factor, .53 to .88 for the *Knowledge of* factor, .55 to .80 for the *Identification with* factor, and .58 to .82 for the *Interaction with* factor. In addition, communalities, or the proportion of variance in the item explained by the factor, ranged from .29 to .77. Table 1 presents the factor loadings and communalities for each item.

To evaluate the convergent validity of the survey and address Research Question 2, we computed the average variance extracted (AVE) for each factor. Convergent validity can be established if both the AVE and standardized factor loadings for all the items are equal to or greater than 0.5 [27]. Both the AVE and standardized factor loadings were above this threshold for all factors and items in the model, suggesting strong convergent validity for the survey, see

**Table 1. Factor loadings and communalities ($h^2$) for Items in PRISM survey.**

| Item | Factor Loading | $h^2$ |
|---|---|---|
| **Interest In** | | |
| I look forward to seeing [NAME]'s content. | .81 | .657 |
| I like to see content that [NAME] has made. | .81 | .656 |
| I hope [NAME] achieves their goals. | .77 | .587 |
| I care about what happens to [NAME]. | .75 | .560 |
| I am interested in [NAME]. | .72 | .520 |
| I would follow [NAME] on another account if they create one on the same social media site that I already follow them on. | .66 | .435 |
| I like to see content that [NAME] is in that they did not make themselves. | .56 | .311 |
| **Knowledge Of** | | |
| I seek out information to learn more about [NAME]. | .88 | .774 |
| Learning about [NAME] is important to me. | .86 | .731 |
| When I come across information about [NAME] I will search to learn more about them. | .85 | .728 |
| If I saw a story about [NAME], I would read it. | .71 | .497 |
| I am aware of the personal details of [NAME]'s life. | .53 | .285 |
| **Identification With** | | |
| I have many of the same beliefs as [NAME]. | .80 | .639 |
| I have many of the same opinions as [NAME]. | .79 | .620 |
| I usually agree with [NAME]. | .76 | .570 |
| I can relate to [NAME]'s attitudes. | .74 | .553 |
| I usually make the same choices as [NAME]. | .67 | .443 |
| I like the way [NAME] handles problems. | .55 | .301 |
| **Interaction With** | | |
| [NAME] makes me feel as if I am with a friend. | .82 | .666 |
| I can imagine myself as [NAME]'s friend. | .75 | .556 |
| [NAME] keeps me company when viewing their content. | .70 | .490 |
| [NAME] understands the kinds of content I want to see. | .58 | .335 |

**Table 2.  Average variance extracted and McDonald's $\omega_t$ internal consistency estimates, standard errors, and confidence intervals by factors.**

| | | | | $\omega_t$ CI 95% | |
|---|---|---|---|---|---|
| Factor | AVE | $\omega_t$ | S.E. | Lower | Upper |
| Interest In | .532 | .789 | .021 | .748 | .828 |
| Knowledge Of | .603 | .853 | .011 | .829 | .872 |
| Identification With | .521 | .827 | .016 | .795 | .857 |
| Interaction With | .512 | .783 | .015 | .750 | .810 |

Note: Bias-corrected and accelerated (BCa) bootstrap CI with 1000 bootstrap iterations

Table 2 for AVE values. According to Cheung and Wang [27], discriminant validity can be established if the correlation between any two factors is below 0.7. This criterion ensures that the factors, while related, are distinct from each other and measure different constructs. Table 3 shows that none of the correlations between the factors in the model were greater than this threshold, suggesting evidence for discriminant validity for the four factors.

We also examined the convergent and discriminant validity of the PRISM by correlating the factor scores with participants' frequency of interaction with the celebrity's online content, defined as engaging with or viewing content that involved the celebrity, and the year they started following the celebrity. A positive correlation between PRISM measures and frequency of interaction would provide evidence of convergent validity, as more frequent interactions are expected to foster stronger parasocial relationships. Our findings reveal a statistically significant positive relationship (Table 4), supporting the convergent validity of the PRISM survey. In other words, participants who reported higher levels of interaction with the celebrity's online content also demonstrated stronger parasocial relationships with that celebrity. This evidence supports the claim that the PRISM survey accurately captures the intended construct of parasocial relationships by aligning with participants' actual engagement with the celebrity's content.

On the other hand, discriminant validity is evidenced when a measure does not associate strongly with other measures that it theoretically should not be associated with. For instance, research [2, 5, 23] has found that factors such as intimacy, credibility, and frequency of interactions are more important at developing parasocial relationships than the duration of time someone has been following a celebrity. Therefore, discriminant validity was evidenced by a non-significant relationship between parasocial relationships, as defined by PRISM, and the length of time following the celebrity. The non-statistically significant correlations between PRISM measures and length of time following the celebrity indicates that the PRISM measures are capturing a distinct concept that is independent of how long someone has been following the celebrity.

To further strengthen the construct validity of the survey, we calculated the internal consistency reliability for each factor. Internal consistency reliability assesses the extent to which the

**Table 3.  Standardized factor correlations.**

| | Interest In | Knowledge Of | Identification With | Interaction With |
|---|---|---|---|---|
| Interest In | 1.000*** | | | |
| Knowledge Of | .607*** | 1.000*** | | |
| Identification With | .654*** | .425*** | 1.000*** | |
| Interaction With | .520*** | .640*** | .655*** | 1.000*** |

***$p < .001$

**Table 4. Correlations between factors and time following celebrity and frequency of interaction.**

| | On average, how often do you interact with [NAME]'s posts online?[a] | In what year did you first start following [NAME]? |
|---|---|---|
| Interest In | .296*** | -.037 |
| Knowledge Of | .251*** | -.081 |
| Identification With | .086* | -.017 |
| Interaction With | .265*** | -.057 |

Notes

*p < .05

**p < .01

***p < .001 (two-tailed test)

[a] Response options: more than once a day, daily, 3 to 5 days per week, once a week, once every two weeks, once a month, 3–5 times a year, twice a year or less

items within each factor are consistently measuring the same underlying construct. The internal consistency reliability was measured using the $\omega_t$ coefficient [38]. The $\omega_t$ coefficient has been shown to be a more sensible measure of internal consistency than Cronbach's alpha, since it does not assume Tau equivalency (i.e., equal factor loadings) among the items as Cronbach's alpha does, and using it along with confidence intervals allows researchers to report a range of values for the internal consistency with some degree of confidence [39]. All of the factors had $\omega_t$ values of .78 or greater, indicating strong internal consistency, see Table 2 for values and confidence intervals.

Finally, to address Research Question 3, we utilized a confirmatory factor analysis (CFA) measurement invariance approach to determine whether the factor structure remained invariant across different social media platforms (Twitch, YouTube). We estimated the multi-group CFA with WLSMV estimation and theta parameterization using the lavaan package in R. For this analysis, we compared YouTube and Twitch as exemplars of recorded and live format social media platforms. We first specified a configural invariance model to examine whether the same factor structure held separately for the YouTube and Twitch social media platforms. This model included our four correlated factors (*Interest In*, *Knowledge Of*, *Identification With*, and *Interaction With*) across the two social media platforms, such that the eight factors were allowed to correlate freely. These factors were estimated simultaneously with factor means and variances set to 0 and 1, respectively. The factor loadings and threshold parameters were all allowed to freely estimated. Theta parameterization, in which the residual variance of each item is fixed to 1, was used for identification purposes. The configural model displayed adequate fit, CFI = .943, TLI = .936, RMSEA = .071, 90% $\text{CI}_{\text{RMSEA}}$ (.067, .075), SRMR = .066 and indicates that the same factor structure holds across both social media platforms.

Next, we assessed the metric invariance model by adding additional parameter constraints to the configural model. In the metric invariance model, the unstandardized factor loadings were constrained to be equal across both platforms. If the fit of this model is similar to the configural model, it indicates that the factors have the same meaning across platforms. Threshold parameters were allowed to freely estimate as was the case in the configural model. The metric invariance model displayed good overall fit, CFI = .950, TLI = .946, RMSEA = .065, 90% $\text{CI}_{\text{RMSEA}}$ (.061, .069), SRMR = .068, and did not display significantly worse fit when compared to the configural model, $\Delta\chi^2 = 61.8$, $p = .202$, $\Delta$CFI = .007, $\Delta$RMSEA = .006, see Table 5 for the model comparisons. These results demonstrate that metric invariance held, indicating that the factors have the same meaning across both platforms.

**Table 5. Model comparisons for validation of measurement invariance.**

| Model | $\chi^2$ | *Df* | CFI | TLI | RMSEA | $\Delta\chi^2$ | $\Delta df$ | *p*-value | ΔCFI | ΔRMSEA |
|---|---|---|---|---|---|---|---|---|---|---|
| Configural | 1355.1 | 406 | .943 | .936 | .071 | | | | | |
| Metric | 1416.9 | 424 | .950 | .946 | .065 | 61.8 | 18 | .202 | .007 | .006 |
| Scalar | 1475.2 | 483 | .947 | .949 | .063 | 58.3 | 59 | .087 | .003 | .002 |

Note: $\chi^2$, scaled chi-squared difference test; CFI, comparative fit index; TLI, Tucker-Lewis index; RMSEA, root mean square error of approximation.

Building off the metric invariance model, we further constrained the threshold parameters to be equal across both platforms in order to test for scalar invariance. Negligible worsening of fit compared with the metric invariance model indicates that the factors have the same meaning for both platforms and participants use the response scale in a similar fashion (e.g., individuals in each group with the same level of the latent factor would have the same score on the indicator variable). The scalar invariance model displayed good overall fit, CFI = .947, TLI = .949, RMSEA = .063, 90% CI$_{RMSEA}$ (.060, .067), SRMR = .066, and did not display significantly worse fit when compared to the metric invariance model, $\Delta\chi^2$ = 58.3, *p* = .087, ΔCFI = .003, ΔRMSEA = .002, see Table 5 for the model comparisons. These results show that scalar invariance held, meaning that both the factor loadings and the thresholds were the same across the two social media platforms. Finally, given the unequal sample sizes in the Twitch and YouTube groups, we ran a sensitivity analysis. We implemented an approach suggested by Yoon & Lai [40] in which a random sample of 108 respondents from the larger group (i.e., YouTube) were taken and measurement invariance was assessed between the groups. We repeated this process 100 times, as recommended by Yoon and Lai, and in every occurrence, conclusions for measurement invariance were confirmed.

## Discussion

In response to the need for a suitable survey to measure parasocial relationships developed through social media, Boyd et al. [11] developed the Parasocial Relationships in Social Media (PRISM) Survey. In the current study, we build on the initial validation work conducted by Boyd et al. by further examining the construct validity of the survey and assessing its measurement invariance across various social media platforms. This comprehensive analysis allows us to strengthen the evidence supporting the validity and reliability of the PRISM Survey for measuring parasocial relationships in the context of social media. The theoretical significance of our study lies in its contribution to the understanding of parasocial relationships with online, social media celebrities. Our findings demonstrated the PRISM survey's ability to replicate and measure four factors related to parasocial relationships (Interest In, Knowledge Of, Identification With, and Interaction With) when developed with a celebrity through social media, and that the relationship is invariant across different social media platforms (e.g., Twitch, YouTube).

Based on the criteria proposed by Cheung and Wang [27] (i.e., both the AVE and standardized factor loadings should exceed 0.50), our findings provide support for the convergent validity of the PRISM Survey. The AVE values and standardized factor loadings obtained from our analysis meet or exceed the recommended threshold, indicating that the items within each factor are strongly related to their respective latent constructs. This further confirms that the PRISM survey effectively measures the intended aspects of parasocial relationships in the context of social media. Further support for the survey's overall reliability was found using the internal consistency coefficient ($\omega_t$). All $\omega_t$ values were .78 or greater, and support that the items have strong internal consistency reliability. Cheung and Wang [27] also state that

constructs have discriminant validity when the correlation between any two factors is less than 0.70. Using this criterion, we were also able to support the claim that the PRISM Survey possesses discriminant validity. This means that the factors, while related, measure distinct aspects of parasocial relationships. Together, these results further support the findings of the original study, indicating that the PRISM survey can be used to obtain reliable and valid data on the parasocial relationships that people develop with celebrities through social media.

Convergent and discriminant validity were also assessed by correlating the factors in the CFA model with a question on the survey that asked respondents what year they started following the celebrity and how often they interact with the celebrity's online content. Findings mirror results from Boyd et al. [11] who found similar convergent and discriminant correlations in their sample. Interestingly, the correlations observed between the PRISM factors and the frequency of interaction with a celebrity's posts were stronger in the current study compared to the findings reported in Boyd et al. [11]. This finding provides additional support for the convergent validity of the PRISM Survey, indicating that higher levels of engagement with a social media celebrity's content are indeed associated with stronger parasocial relationships.

The "Identification With" dimension exhibited the weakest correlation with the frequency of respondents' interactions with the celebrity's online posts. The low correlation indicates that the degree of identification with a social media celebrity is not predominantly influenced by how often a person interacts with a celebrity's content. The "Identification With" factor captures the extent to which the respondent feels similar to, or shares the values and goals of, the social media celebrity. Thus, identification depends more on other factors, such as the similarity of beliefs, opinions, attitudes, and choices rather than the simple frequency of interactions. It is also possible that identification with a social media celebrity may have a nonlinear relationship with the frequency of interaction with their content. For example, it may be that identification only increases significantly after a certain level of interaction is reached, or that identification reaches a plateau or decreases after too much interaction. Future research could test this hypothesis by using different measures of interaction frequency, such as the duration or intensity of interaction. Finally, the low correlation does not necessarily imply that the "Identification With" scale is invalid or unreliable. The scale has shown adequate content and construct validity, as well as internal consistency, in both the current study and the previous study by Boyd et al. [11]. The scale also has high factor loadings and average variance extracted, indicating that the items measure the same underlying construct of "Identification With" a social media celebrity.

The lack of significant correlations between the PRISM factors and the year a respondent began following the celebrity provides additional discriminant validity evidence. These results suggest that the PRISM factors are not simply a function of the duration of time an individual has been following a celebrity, highlighting that parasocial relationships can develop even after a relatively brief interaction. The significant positive correlations between the PRISM factors and how often a respondent interacts with the celebrity's online content provide convergent validity evidence that while more engagement with the social media celebrity is associated with stronger parasocial relationships, they are not solely determined by frequency of interaction, allowing parasocial relationships to develop even with lower levels of engagement. The analysis showed that the survey had full scalar invariance between the two social media platforms (Twitch and YouTube), meaning the scale was measuring the same constructs across both platforms. These findings have significant implications for research on parasocial relationships. YouTube and Twitch use very different formats in the media they produce. YouTube is primarily used for uploading prerecorded videos while Twitch uses a live stream format that viewers watch in real time. By successfully finding scalar measurement invariance between these two platforms, we can extrapolate that the PRISM survey might behave similarly when

used with other social media platforms that use the prerecorded and live formats as well (e.g., Dtube, Vimeo). This allows researchers to make meaningful comparisons of the parasocial relationships developed through different platforms when measuring them with the PRISM survey. This includes research on the effectiveness of different platforms in fostering these relationships terms of strength, speed, and longevity.

The PRISM survey is intended to be used to measure the parasocial relationships that people develop with online celebrities through social media, such as YouTube, Twitch, Twitter, and Facebook. By developing valid measures of this phenomenon, researchers can more effectively analyze its effects on other phenomena, and more importantly, people themselves. Using this survey, researchers can easily expand parasocial relationship research into the realm of social media and continue researching its impacts on marketing and brand perceptions [2–4], political support [41], attachment styles [42, 43], loneliness [8], and more. For example, managers for social media celebrities could use PRISM to assess the quality and strength of the parasocial relationships their clients have with their audiences. This could help them design more engaging and satisfying media content, attract more followers, and help foster more loyal and active fan communities. Educators could use PRISM as a pedagogical tool to teach students about the psychological and social aspects of media consumption as well as promote media literacy. Policymakers could also use the PRISM survey as a tool to monitor the potential benefits and harms of parasocial relationships, such as their impact on mental health and civic engagement. The survey also allows researchers to address questions about the impacts of these relationships with social media personalities and how they impact our behaviors and opinions. Do these relationships create unhealthy attachments with celebrities, or do they allow us other avenues for interaction and intimacy when face to face relationships might otherwise be unavailable? This is especially pertinent with the effects of the COVID-19 pandemic still impacting day to day life and interactions of many.

While the study provides additional evidence on the construct validity of the PRISM survey, it is not without its limitations. The sample size used was modest (n = 606) and consisted of mostly young, American adults (mean age 21.5 years) who were recruited online from various social media platforms. In addition, even though the data was collected at an international level, the sample was limited to those who could read English. Therefore, the generalizability of the findings may be limited. Future research should strive for a larger and more diverse sample. In addition, large imbalances in the group sizes may affect the results of the invariance analysis. While we ran sensitivity analyses to gauge the magnitude of this issue, future research should investigate more social media platforms and strive for a more balanced design for these comparisons. The instrument also relies on self-reported measures, which may be subject to social desirability bias, memory bias, or lack of retrospection. The study used a cross-sectional design which does not allow us to track or gauge the relationships over time.

Despite these limitations, the study lays the groundwork for numerous practical applications and uses. The survey results can be examined at the level of the individual items or factors and comparisons can be made across different social media platforms. Researchers are encouraged to continue validating the survey with additional social media platforms, such as Discord, Facebook, Twitter, and TikTok as well as across diverse samples of participants. This includes underrepresented populations such as ethnic minorities, members of the LGBTQ+ community, and other marginalized populations. Future research should also be dedicated to translating and validating the PRISM survey in other languages, as the current study was only conducted in English. Translation into other languages would allow the survey to be further assessed for measurement invariance to determine if different cultures, which have already been shown to experience parasocial relationships [2, 4, 21, 44–46], experience them in the same way.

The PRISM survey offers a novel and psychometrically sound method for researchers to measure parasocial relationships that have been formed through social media. The survey demonstrated strong reliability, construct validity, and measurement invariance across different platforms. The PRISM Survey can be used to assess the intensity and breadth of parasocial relationships with online celebrities, as well as the factors that influence them. The survey can also facilitate cross-platform comparisons of parasocial phenomena and their effects on various outcomes, such as marketing, brand support, attachment, and loneliness. The PRISM Survey contributes to the advancement of parasocial relationship research in the social media domain and opens up new avenues for exploring the role of online celebrities in shaping people's attitudes, behaviors, and well-being.

## Author Contributions

**Conceptualization:** Austin T. Boyd, Louis M. Rocconi.

**Formal analysis:** Austin T. Boyd.

**Methodology:** Austin T. Boyd, Louis M. Rocconi.

**Project administration:** Louis M. Rocconi, Jennifer Ann Morrow.

**Supervision:** Louis M. Rocconi.

**Writing – original draft:** Austin T. Boyd.

**Writing – review & editing:** Austin T. Boyd, Louis M. Rocconi, Jennifer Ann Morrow.

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
