## [Decision Letter · Decision Letter 0]

24 Jan 2024

PONE-D-23-12810Construct Validation and Measurement Invariance of the Parasocial Relationships in Social Media SurveyPLOS ONE

Dear Dr. Rocconi,

Thank you for submitting your manuscript to PLOS ONE. After careful consideration, we feel that it has merit but does not fully meet PLOS ONE’s publication criteria as it currently stands. Therefore, we invite you to submit a revised version of the manuscript that addresses the points raised during the review process esp. by the second reviewer.

We look forward to receiving your revised manuscript.

Kind regards,

Frantisek Sudzina

Academic Editor

PLOS ONE

Journal Requirements:

Reviewers' comments:

Reviewer's Responses to Questions

**Comments to the Author**

1. Is the manuscript technically sound, and do the data support the conclusions?

Reviewer #1: Yes

Reviewer #2: Partly

2. Has the statistical analysis been performed appropriately and rigorously? 

Reviewer #1: I Don't Know

Reviewer #2: Yes

3. Have the authors made all data underlying the findings in their manuscript fully available?

Reviewer #1: Yes

Reviewer #2: No

4. Is the manuscript presented in an intelligible fashion and written in standard English?

Reviewer #1: Yes

Reviewer #2: Yes

5. Review Comments to the Author

Reviewer #1: Review of PONE-D-23-12810

This study is an examination of the psychometric properties of the Parasocial Relationships in Social Media (PRISM) scale. As such, the authors examine important components of a sound scale, including replication of the factor structure, convergent and discriminant validity, and measurement invariance. Some of the statistics used to evaluate the survey are beyond my expertise, but the authors explained most of them clearly and demonstrated the evidence that the measure is sound.

My major critique of this paper concerns the elements of the scale itself, in that they are not indicative of a relationship so much as an attitude of interest toward a celebrity. Only the items in the “Interaction With” scale (ironically, given the discussion in the literature regarding the difference between parasocial interactions and relationships), tap into the socioemotional basis of a relationship, and they are limited to friendship. This measure does not seem to allow for participants whose relationship with a media figure is romantic or that of a mentor/mentee (or anything other than friendship). In that sense, the notion that this survey measures PSR per se is as flawed as the measures (PSI, API) from which items were derived. If these relationships can “feel as real to viewers as real-life relationships” (p. 3) then why aren’t any of the items focused on identifying the social and emotional affordances of real relationships that are available in parasocial relationships? This scale appears to me not to measure an imagined relationship so much as a set of behaviors and attitudes connected to a media figure.

The introduction provides a rationale for the need for the scale as well as the study’s approach to examining its properties. The review section on Parasocial Relationships is also appropriate for this context, as is the description of previous work on the PRISM survey.

The method was a little odd in not starting with a description of the sample and in focusing on describing the analyses conducted, which I think of as information that typically appears in Results. Perhaps such a format is standard for measurement papers.

In the Results section, I was a little confused as to why a lack of correlation between the PRISM subscales and the year in which the participant first started following the celebrity was considered evidence of discriminant validity. Relationship models (for real relationships) emphasize not just frequency of interactions, but the duration of the relationship. Why wouldn’t the same be true for parasocial relationships?

Was “interact with” defined for participants? I’m not sure if it includes simply reading/watching the media figure’s posts or if, at minimum, the participant had to respond to the post in some way (like it) for that to count as an interaction—or perhaps it was left open to interpretation by the participant?

I might not fully understand the section on measurement invariance but if the sample was separated by platform used (YouTube v. Twitch), how would that work with so few people reporting that they followed their media figures on Twitch? Isn’t this comparison rather unbalanced?

Small things:

The second paragraph of the introduction in the section on Parasocial Relationships is enormous and covers multiple ideas. I suggest that it be split up. The end of this paragraph also states that the PRISM survey does not need to be modified for use in social media contexts, but it raised the question for me—why create a scale for use with social media rather than one that could be used for any context? Is that not possible? Does each context thus need its own survey instrument?

In Results, descriptions of the information presented in the tables should be adjacent to the tables themselves (e.g., discussion of measures of internal consistency, presented in table 2, came two pages later after table 4).

I noted a few typos throughout the manuscript.

Reviewer #2: 1.Currently, there are some establised PSI scales that have been widely adopted with high levels of reliability and validity, and some of them are developed within social media context, e.g. Dibble, J. L., Hartmann, T., & Rosaen, S. F. (2016). Parasocial interaction and parasocial relationship: Conceptual clarification and a critical assessment of measures . Regrettably, the introductory section lacks a clear delineation of the distinctive aspects of the authors' study and fails to elucidate the necessity for the re-creation of the scale.

2. While the literature review has touched upon differences between social media and traditional media, such as interactivity, the justification for the necessity of revising the scale remains insufficient. Further clarification is required regarding the theoretical significance and practical application value of this scale revision.

3.The demographic data for the sample should be provided in detail, outlining key demographic characteristics. Additionally, clear criteria for the selection of the 227 celebrities need to be explicitly stated.

4. The correlation of the Identification With dimension is notably low. Therefore, the need for revision of this dimension is a matter that requires further consideration in the data analysis.

5. The discussion section needs a more comprehensive elucidation of the theoretical significance and practical value of this study. Furthermore, limitations of the research and future prospects are currently lacking in sufficient detail.

6. PLOS authors have the option to publish the peer review history of their article (what does this mean?). If published, this will include your full peer review and any attached files.

Reviewer #1: No

Reviewer #2: No

---

## [Author Response · Author response to Decision Letter 0]

18 Feb 2024

Dear Dr. Sudzina,

We would like to thank you and the reviewers for your insightful comments on our manuscript and for the opportunity to improve our manuscript. We appreciate the reviewers’ acknowledgment of the merits of the study and are especially grateful for the important questions raised and areas of improvement identified in the comments. As a result, we have incorporated the reviewers’ recommendations in our revised manuscript. To ease the review process, we have listed the reviewers’ comments and a corresponding narrative describing our response. Please see the attached file which details our revisions. We look forward to your thoughts on our revisions. We hope you agree that the manuscript is now in a better position to be informative and valuable to the PLOS One readership.

Sincerely,

The authors

---

## [Editor Report · Decision Letter 1]

27 Feb 2024

Construct Validation and Measurement Invariance of the Parasocial Relationships in Social Media Survey

PONE-D-23-12810R1

Dear Dr. Rocconi,

We’re pleased to inform you that your manuscript has been judged scientifically suitable for publication and will be formally accepted for publication once it meets all outstanding technical requirements.

Kind regards,

Frantisek Sudzina

Academic Editor

PLOS ONE
---

## [Editor Report · Acceptance letter]

19 Mar 2024

PONE-D-23-12810R1 

PLOS ONE

Dear Dr. Rocconi, 

I'm pleased to inform you that your manuscript has been deemed suitable for publication in PLOS ONE. Congratulations! Your manuscript is now being handed over to our production team.

Kind regards, 

on behalf of

Dr. Frantisek Sudzina 

Academic Editor

PLOS ONE